# Metabolic phenotyping by treatment modality in obese women with gestational diabetes suggests diverse pathophysiology: An exploratory study

Sara L. White[1]*, Shahina Begum[1], Matias C. Vieira[1], Paul Seed[1], Deborah L. Lawlor[2,3,4], Naveed Sattar[5], Scott M. Nelson[6], Paul Welsh[5], Dharmintra Pasupathy[1], Lucilla Poston[1], on behalf of UPBEAT Consortium[1¶]

1 Department of Women and Children's Health, School of Life Course Sciences, King's College London, London, United Kingdom, 2 MRC Integrative Epidemiology Unit at the University of Bristol, Bristol, United Kingdom, 3 Population Health Science, Bristol Medical School, University of Bristol, Bristol, United Kingdom, 4 NIHR Bristol Biomedical Research Centre at University Hospitals Bristol NHS Foundation Trust and University of Bristol, Bristol, United Kingdom, 5 Institute of Cardiovascular and Medical Sciences, University of Glasgow, Glasgow, United Kingdom, 6 School of Medicine, University of Glasgow, Glasgow, United Kingdom

¶ Membership of the UPBEAT Consortium is provided in the Acknowledgments.
* sara.white@kcl.ac.uk

**Data Availability Statement:** Due to the limitations of the consent provided by the patients in our study, and restrictions imposed by our funders we

## Abstract

### Background and purpose

Excess insulin resistance is considered the predominant pathophysiological mechanism in obese women who develop gestational diabetes (GDM). We hypothesised that obese women requiring differing treatment modalities for GDM may have diverse underlying metabolic pathways.

### Methods

In this secondary analysis of the UK pregnancies Better Eating and Activity Trial (UPBEAT) we studied women from the control arm with complete biochemical data at three gestational time points; at $15–18^{+6}$ and $27–28^{+6}$ weeks (before treatment), and $34–36^{+0}$ weeks (after treatment). A total of 89 analytes were measured (plasma/serum) using a targeted nuclear magnetic resonance (NMR) platform and conventional assays. We used linear regression with appropriate adjustment to model metabolite concentration, stratified by treatment group.

### Main findings

300 women (median BMI 35kg/m$^2$; inter quartile range 32.8–38.2) were studied. 71 developed GDM; 28 received dietary treatment only, 20 metformin, and 23 received insulin. Prior to the initiation of treatment, multiple metabolites differed ($p<0.05$) between the diet and insulin-treated groups, especially very large density lipoprotein (VLDL) and high density lipoprotein (HDL) subclasses and constituents, with some differences maintained at 34–36

cannot make the data generally available. The UPBEAT Scientific Advisory Committee accept applications for use of data from those who make a formal request, providing a description of the intended study on a research application form (UPBEAT RAF) available from Glen Nishku (glen. nishku@gstt.nhs.uk). Providing the proposed studies do not conflict with consent, the data will be freely available.

**Funding:** This study received funding from the National Institute of Health Research (www.nihr.ac. uk) (RP-PG-0407-10452), Medical Research Council UK (www.mrc.ac.uk) (MR/ L002477/1), Chief Scientist Office, Scottish Government Health Directorates (Edinburgh) (www.cso.scot.nhs.uk) (CZB/A/680), Biomedical Research Centre at Guys & St Thomas NHS Foundation Trust & King's College London (www.guysandstthomasbrc.nihr. ac.uk) and the NIHR Bristol Biomedical Research Centre (www.bristolbrc.nihr.ac.uk), Tommy's Charity, UK (www.tommys.org) (SC039280). SLW was supported by a fellowship from Diabetes UK (www.diabetes.org.uk) (14/ 0004849). DP was funded by Tommy's Charity. MCV was supported by a fellowship from CAPES-Brazil (www.iie.org/ programs/CAPES) (BEX 9571/13-2). DAL's contribution to this work was supported by the European Union's Seventh Framework Programme (www.ec.europa.eu/research/fp7) (FP7/2007-2013), ERC grant agreement (www.erc.europa.eu) (No 669545, DevelopObese) and the US National Institute of Health (www.nih.gov) (R01 DK10324) and is a National Institute for Health research Senior Investigator (NF-SI-0166-10196) and LP is an Emeritus National Institute for Health Research Senior Investigator (NI-SI-0512-10104). The study sponsor or funders had no role in study design, data collection and analysis, decision to publish, or preparation of the manuscript.

**Competing interests:** SMN and LP disclose receipt of lecture/other fees from Roche Diagnostics, outside the scope of this work. DAL discloses receipt of fees from Roche Diagnostics and Ferring Pharmaceuticals for research unrelated to this paper. All other authors declare that there is no declaration of interest associated with their contribution to this manuscript. This does not alter our adherence to PLOS ONE policies on sharing data and materials.

weeks' gestation despite treatment. Gestational lipid profiles of the diet treatment group were indicative of a lower insulin resistance profile, when compared to both insulin-treated women and those without GDM. At 28 weeks' the diet treatment group had lower plasma fasting glucose and insulin than women treated with insulin, yet similar to those without GDM, consistent with a glycaemic mechanism independent of insulin resistance.

## Conclusions/Interpretation

This exploratory study suggests that GDM pathophysiological processes may differ amongst obese women who require different treatment modalities to achieve glucose control and can be revealed using metabolic profiling.

## Introduction

For most healthcare professionals and pregnant women, gestational diabetes mellitus (GDM) diagnosis is understood to be a binary categorisation of hyperglycaemia versus normoglycaemia. This is despite a well-established linear increase in risk of adverse outcomes across the glycaemic spectrum [1], and the potential for pathophysiological heterogeneity of GDM with diverse maternal and offspring outcomes [2].

Hyperglycaemia in pregnancy is widely accepted to result from an imbalance between rising insulin resistance and inadequate insulin secretion, yet specific mechanisms likely differ between, and amongst phenotypic groups. Amongst obese women for example, excessive insulin resistance is considered to be the predominant pathophysiological mechanism, whereas insulin secretory defects may predominate in lean women with GDM [2–5]. This distinction was corroborated in a recent study in which biochemical and clinical heterogeneity were described in women with GDM, classified as GDM with an insulin secretion defect, GDM with an insulin sensitivity defect, and mixed defects. The authors reported that women with a predominant insulin sensitivity defect were those with a higher BMI [2].

Health care professionals appreciate that women with GDM will require different treatment modalities according to their ability to control glycaemia. While this may reflect disease severity, maternal lifestyle or adherence to treatment, diverse underlying disease processes may also be contributory. Improved understanding of the pathophysiology could facilitate management through delineation of subtypes of GDM, enabling targeted therapy [6].

We have previously shown that obese women with GDM have differing metabolic profiles from obese women without GDM and that this is evident prior to diagnosis [7]. Using the same dataset, we have hypothesised that the measured analytes might further distinguish between groups necessitating diverse treatment approaches to achieve glucose control. Metabolite phenotypes were therefore compared between women allocated to different GDM treatment strategies in a proof of principle exploratory study. To our knowledge, there has been no previous attempt to define subgroups according to measured analytes, and by treatment strategy.

## Materials and methods

### Study design

This prospective cohort study was a secondary analysis utilising data from the UK Pregnancies Better Eating and Activity Trial (UPBEAT, ISRCTN 89971375), a multicentre RCT of a

complex dietary and physical activity intervention designed to prevent GDM in obese women and reduce the incidence of LGA infants [8]. Women with a pre-existing diagnosis of essential hypertension, diabetes, coeliac disease, thyroid disease, renal disease, systemic lupus erythematosus, antiphospholipid syndrome, sickle-cell disease, thalassaemia, current psychosis, or a current prescription of metformin were excluded. The UPBEAT trial (recruitment 2009 to 2014), included 1555 women; they were >16 years of age, had a Body Mass Index (BMI) $\geq 30 kg/m^2$ and a singleton pregnancy. Women were randomised between $15^{+0}$ and $18^{+6}$ weeks' gestation to either a behavioural intervention superimposed on standard antenatal care or standard antenatal care. All aspects of the trial, including the analyses for the present study were approved by the NHS Research Ethics Committee (UK Integrated Research Application System; reference 09/H0802/5) and all participants, including women aged 16 and 17 using Fraser guidelines, provided informed written consent [8].

## Participants

A complete-case analysis was undertaken and included all women from the control arm of UPBEAT who had undertaken a diagnostic Oral Glucose Tolerance Test (OGTT), with documented GDM treatment modality and complete biochemical data at trial entry, at the time of GDM testing and in late pregnancy (*n* = 300). Women were excluded if these criteria were not met, if GDM was diagnosed by local thresholds but did not fulfil diagnostic criteria according to the trial protocol (*n* = 3), or who fulfilled the trial protocol diagnostic criteria, but not local criteria for GDM diagnosis (*n* = 23).

## Procedures

Sociodemographic and clinical data, and non-fasting blood samples were collected at time point 1 ($15–18^{+6}$ gestational weeks'; mean $17^{+0}$). The trial protocol specified that an OGTT should be performed between 27 and $28^{+6}$ gestational weeks', however a clinically pragmatic approach has been adopted for the purposes of this study with inclusion of OGTTs undertaken between $23^{+3}$ and $29^{+6}$ (mean $27^{+5}$). A research blood sample was collected at the time of the OGTT fasting sample (time point 2). Diagnosis of GDM was according to International Association of Diabetes and Pregnancy Study Groups (IADPSG) criteria (fasting glucose $\geq 5.1$ mmol/l, 1 hr $\geq 10.0$ mmol/l, 2 hr $\geq 8.5$ mmol/l) in response to an oral 75g glucose load [9]. A non-fasting blood sample was collected at time point 3 ($34–36^{+0}$ gestational weeks', mean $34^{+6}$). Pregnancy outcome data was recorded shortly after delivery.

The main outcome of interest was GDM treatment modality following diagnosis. Women were subcategorised into: No-GDM; GDM Diet Group (treated with diet only); GDM Metformin Group (treated with metformin); GDM Insulin Group (treated with insulin alone or metformin plus insulin). Study centres reported that GDM treatment most frequently began with dietary advice, followed by the addition of metformin and then insulin if control was not achieved, either due to glycaemic severity or poor compliance. Modality was recorded as the treatment at the time of delivery.

## Metabolic profiling

Analytes were measured in plasma and serum samples using a combination of NMR spectroscopy and conventional assays. A high-throughput targeted NMR metabolomic platform was utilised (http://www.computationalmedicine.fi/platform). The quantitative NMR measures include numerous lipid species, fatty acids, amino acids, and markers of glucose homeostasis and has been used widely in population-based studies of insulin resistance and metabolic disease [10–14]. The methodology has been described previously [15]. Analytes measured using

conventional laboratory platforms (S1 Table) included glucose homeostasis markers, sex hormone binding globulin (SHBG), gamma glutamyl transferase (gGT) and adiponectin. For the purposes of this study and to restrict multiple comparisons, only those analytes identified previously as different between women with GDM and women without GDM at the time of diagnosis (time point 2) [7] were explored. A total of 89 analytes, 83 from the NMR metabolome, were evaluated.

### Statistical analysis

The distribution of data for each analyte was checked for normality and those with non-parametric distribution log-transformed. Relationships between the concentration of variables and gestational age were explored; none required transformation.

Analyte data at time points 1, 2 and 3 were compared by treatment modality group by multivariate regression analyses, with No-GDM as the baseline group for comparison. Standard Deviation (SD) differences between each treatment category and No-GDM are reported to enable comparison between analytes with differing units of measurement. Exploratory analyses compared women treated by insulin with those treated by diet.

An *a priori* decision based on known associations identified age and BMI as confounders for the multivariate analyses. Each regression was clustered by centre.

Pregnancy outcomes between GDM treatment groups were compared using either one-way ANOVA or the Kruskal Wallis test depending on the distribution of data.

Due to small numbers and the exploratory nature of this investigation, no sensitivity analyses were undertaken, although differences between women with GDM included and those excluded, were investigated. No formal correction for multiple testing was undertaken and statistical significance was assumed at a *p* value <0.05.

Statistical analysis was performed using Stata software, version 14.0 (StataCorp LP, College Station, Texas).

## Results

Of the 664 women in the control arm of UPBEAT, 300 with complete biochemical data were included (median BMI 35 kg/m$^2$, Interquartile range (IQR) 32.8–38.2). Of these, 229 did not develop GDM (No-GDM group); 71 (24%) developed GDM, of whom 28 (39%) were treated by diet (Diet Group); 20 (28%) with metformin (Metformin Group); and 23 (32%) with insulin (Insulin Group; 9 insulin alone, 13 insulin plus metformin) (S1 Fig).

Participant characteristics and pregnancy outcome by treatment modality are summarised in Table 1. Comparison between women with GDM included in this study compared to those excluded are shown in S2 Table.

The analyte profiles are illustrated by a representative subset (*n* = 22) of different metabolite groups (Figs 1–3). Metabolite absolute values at each time point for this subset are shown in S3–S5 Tables. Absolute values and graphical representation for all measured analytes are available on request.

### Analytes by treatment modality

**Diet, metformin and insulin groups.** *Time point 1; 10 weeks before diagnosis/treatment (random blood sample)*. At least 10 weeks before OGTT and initiation of treatment, differences between the metabolic profiles of treatment modality groups were identified (Fig 1, S3 Table). Greater concentrations of total lipids within VLDL were observed in the pharmacologically treated groups (Metformin and Insulin Groups), whereas women in the Diet Group demonstrated VLDL lipid concentrations similar to those who did not develop GDM. The Diet group

**Table 1. Maternal clinical factors and pregnancy outcome by treatment modality in women with and without GDM.**

| | No GDM | GDM | | |
|---|---|---|---|---|
| | (*n* = 229) | Diet (*n* = 28) | Metformin (*n* = 20) | Insulin (*n* = 23) |
| | mean (SD)/ median (IQR) | mean (SD) / median (IQR) | mean (SD) / median (IQR) | mean (SD) / median (IQR) |
| | n (%) | n (%) | n (%) | n (%) |
| **Maternal factors (collected time point 1)** | | | | |
| Age (years) | 30.5 (5.5) | 32.7 (4.8) | 31.6 (5.9) | 32.2 (5.1) |
| **Blood pressure (mmHg)** | | | | |
| Systolic | 118.7 (10.8) | 123.5 (12.5) | 119.2 (7.3) | 118.8 (8.3) |
| Diastolic | 72.7 (7.7) | 75.3 (7.8) | 72.1 (5.8) | 72.8 (6.0) |
| **BMI (kg/m$^2$)** | 34.8 (32.7–37.8) | 35 (32.4–39.5) | 36.1 (33.5–39.9) | 37.4 (34.6–40) |
| **Ethnicity** | | | | |
| African | 20 (8.7) | 9 (32.1) | 3 (15) | 2 (8.7) |
| African Caribbean | 12 (5.2) | 4 (14.3) | 1 (5) | 1 (4.3) |
| South Asian | 17 (7.4) | 1 (3.6) | 2 (10) | 1 (4.3) |
| European | 161 (70.3) | 12 (42.9) | 12 (60) | 16 (69.6) |
| Other | 19 (8.3) | 2 (7.1) | 2 (10) | 3 (13) |
| **Parity** | | | | |
| Nulliparous | 111 (48.5) | 10 (35.7) | 11 (55) | 13 (56.5) |
| **Current Smoking** | 14 (6.1) | 1 (3.6) | 1 (5) | 2 (8.7) |
| **Centre** | | | | |
| St Thomas' Hospital | 62 (27.1) | 19 (67.9) | 12 (60) | 8 (34.8) |
| Newcastle | 43 (18.8) | 4 (14.3) | 4 (20) | 0 (0) |
| Glasgow | 67 (29.3) | 2 (7.1) | 3 (15) | 11 (47.8) |
| Manchester | 24 (10.5) | 1 (3.6) | 0 (0) | 2 (8.7) |
| Bradford | 12 (5.2) | 0 (0) | 0 (0) | 1 (4.3) |
| St Georges Hospital | 21 (9.2) | 2 (7.1) | 1 (5) | 1 (4.3) |
| **OGTT results** | | | | |
| Fasting glucose | 4.5 (0.3) | 4.9 (0.5) | 5.4 (0.6) | 5.4 (0.6) |
| 1hr glucose | 7.4 (1.4) | 9.7 (1.8) | 11.4 (1.5) | 10.7 (1.8) |
| 2hr glucose | 5.5 (1.1) | 6.8 (1.2) | 7.5 (1.2) | 7.4 (1.8) |
| **Gestational age (weeks)** | | | | |
| Time point 1 | 16.9 (1.1) | 17.2 (1.1) | 17.1 (1.0) | 17.5 (0.9) |
| Time point 2 | 27.7 (0.7) | 27.8 (0.6) | 27.9 (0.6) | 27.8 (0.5) |
| Time point 3 | 34.9 (0.8) | 34.7 (0.5) | 34.7 (0.5) | 34.5 (0.5) |
| **Pregnancy outcomes** | | | | |
| Preeclampsia | 8 (3.5) | 2 (7.1) | 1 (5) | 2 (9.1) |
| PPH | 34 (14.8) | 5 (17.9) | 2 (10) | 6 (26.1) |
| Weight change (Kg)* | 2.7 (2.1) | 1.0 (2.4) | -1.0 (2.0) | 1.0 (2.6) |
| NICU | 7 (3.1) | 4 (14.3) | 0 (0) | 3 (13) |
| Apgar <7 at 1 min | 3 (1.3) | 1 (3.6) | 0 (0) | 1 (4.3) |
| Preterm birth | 4 (1.7) | 1 (3.6) | 1 (5) | 0 (0) |
| CS all | 77 (33.6) | 11 (39.3) | 6 (30) | 11 (47.8) |
| CS emergency | 45 (19.7) | 7 (25) | 4 (20) | 5 (21.7) |
| GA at delivery | 40.6 (39.3–41.4) | 39.8 (38.8–40.6) | 38.6 (38.1–39.2) | 38.3 (37.9–38.6) |
| Birthweight (g) | 3545.4 (487.9) | 3386.4 (556.1) | 3288.5 (360.1) | 3314.6 (379.4) |
| SGA (customised centile) | 23 (10) | 5 (17.9) | 1 (5) | 1 (4.3) |
| BW (customised centile) | 44.1 (22.8–68.1) | 51.5 (20.9–76.2) | 53.2 (30.3–76.3) | 64.1 (40.7–81.5) |

(*Continued*)

**Table 1.** (Continued)

| | No GDM | GDM | | |
|---|---|---|---|---|
| | (*n* = 229) | Diet (*n* = 28) | Metformin (*n* = 20) | Insulin (*n* = 23) |
| | mean (SD)/ median (IQR) | mean (SD) / median (IQR) | mean (SD) / median (IQR) | mean (SD) / median (IQR) |
| | n (%) | n (%) | n (%) | n (%) |
| LGA (customised) | 15 (6.6) | 3 (10.7) | 1 (5) | 3 (13) |

*weight change between time point 2 and 3, GDM gestational diabetes, SD standard deviation, IQR interquartile range, OGTT oral glucose tolerance test, PPH post-partum haemorrhage, NICU neonatal intensive care, CS caesarean section, GA gestational age, SGA small for gestational age, BW birthweight, LGA large for gestational age. Time point 1—mean $17^{+0}$ weeks, time point 2—mean $27^{+5}$ weeks, time point 3—mean $34^{+6}$ weeks, missing data: systolic and diastolic blood pressure—5, Apgar—2, 1hr glucose—16

had larger large density lipoprotein (LDL) and HDL particles compared to No-GDM women, whereas the pharmacologically treated groups had larger VLDL particles. Women in the Metformin Group had lower polyunsaturated fatty acid: total fatty acid ratios (PUFA:TFA) and

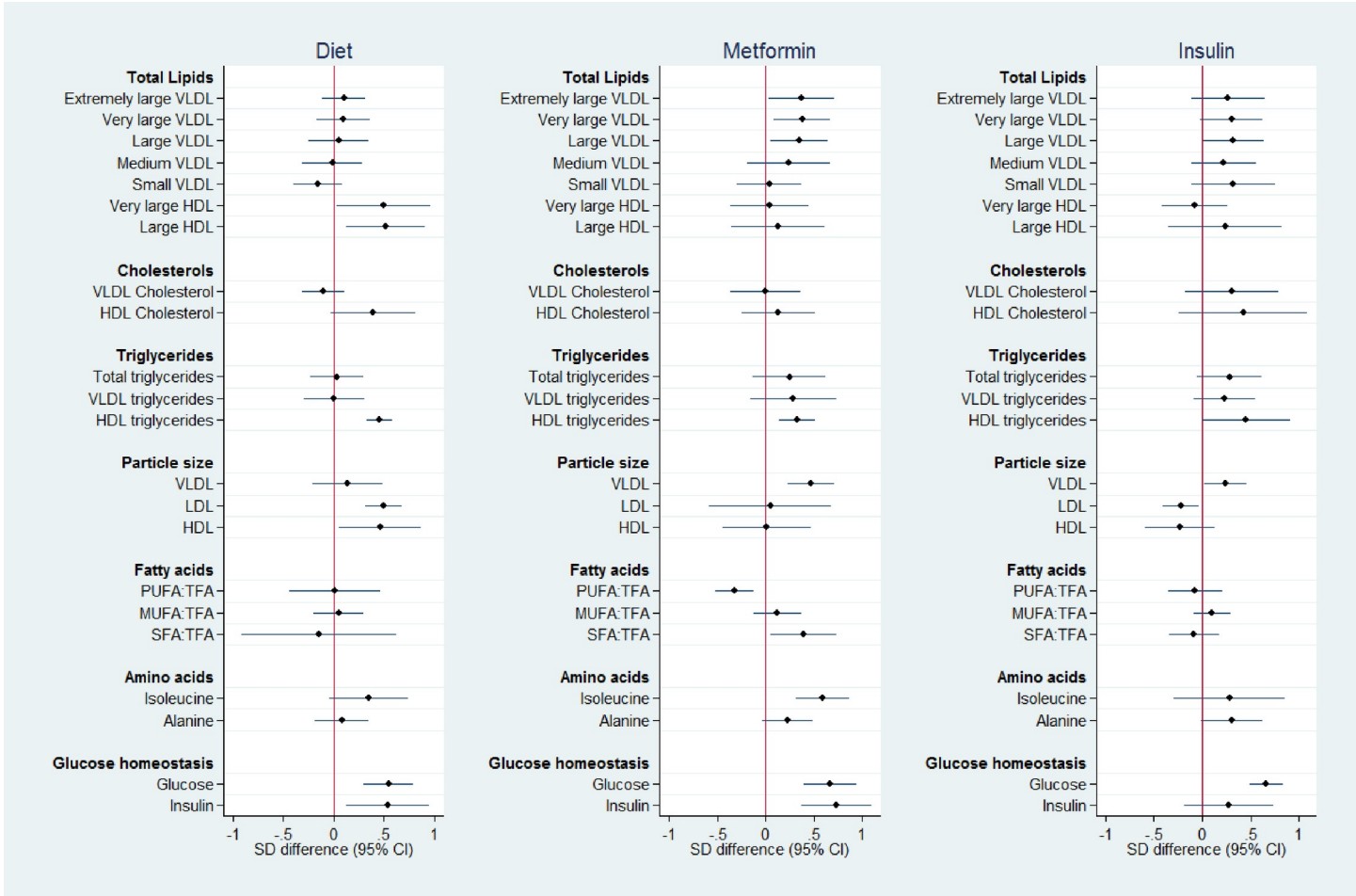

**Fig 1. Metabolite SD difference between GDM treatment groups compared to No-GDM women at time point 1, 10 weeks before diagnosis/treatment (mean $17^{+0}$ weeks').** Data points show the standard deviation (SD) difference between treatment group and No-GDM women. Positive differences compared to No-GDM are shown to the right, negative to the left. PUFA:TFA polyunsaturated fatty acids to total fatty acid ratio, MUFA:TFA monounsaturated fatty acid to total fatty acid ratio, SFA:TFA saturated fatty acid to total fatty acid ratio.

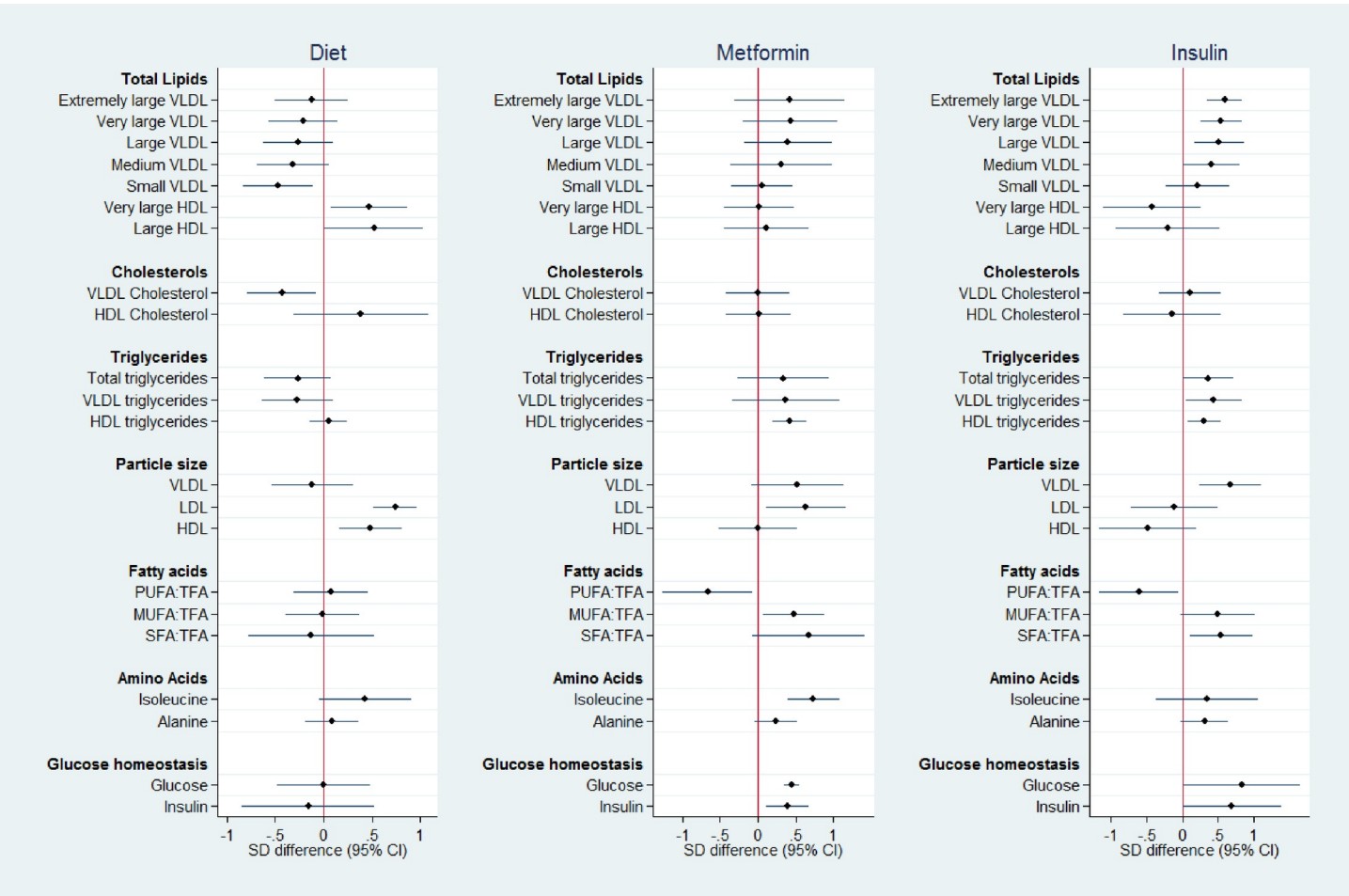

**Fig 2. Metabolite SD difference between GDM treatment groups compared to No-GDM women at time point 2, at time of OGTT (mean 27$^{+5}$ weeks').** Data points show the standard deviation (SD) difference between treatment group and No-GDM women. Positive differences compared to No-GDM are shown to the right, negative to the left. PUFA:TFA polyunsaturated fatty acids to total fatty acid ratio, MUFA:TFA monounsaturated fatty acid to total fatty acid ratio, SFA:TFA saturated fatty acid to total fatty acid ratio.

higher saturated fatty acid:total fatty acid ratios (SFA:TFA) than No-GDM women. The branched-chain amino acid isoleucine was higher in women in the Metformin Group than in women without GDM. Non-fasting glucose was higher than No-GDM in Diet, Metformin and Insulin Groups. Amongst the women eventually treated with insulin, the non-fasting insulin concentration varied widely and was no different from No-GDM women. This contrasted with a significantly higher insulin pre-treatment concentration in the Diet and Metformin Groups than in women without GDM (Fig 1)

*Time point 2; at OGTT (fasting blood sample).* At time point 2, the time of GDM diagnosis but prior to treatment initiation, divergence in analytes between the groups had widened (Fig 2, S4 Table). The Insulin Group had higher concentrations of total lipids in most VLDL subclasses than women without GDM. A similar trend was seen in the Metformin Group. Women in the Diet Group, in contrast, had lower total lipids in small VLDL, higher total lipids in very large and large HDL subclasses and lower VLDL cholesterols compared to women without GDM. VLDL particle size, total triglycerides and triglycerides in both VLDL and HDL were greater in the Insulin Group than women without GDM. In the Diet Group HDL and LDL particle size were greater. The PUFA:TFA ratio was lower in both the metformin and insulin

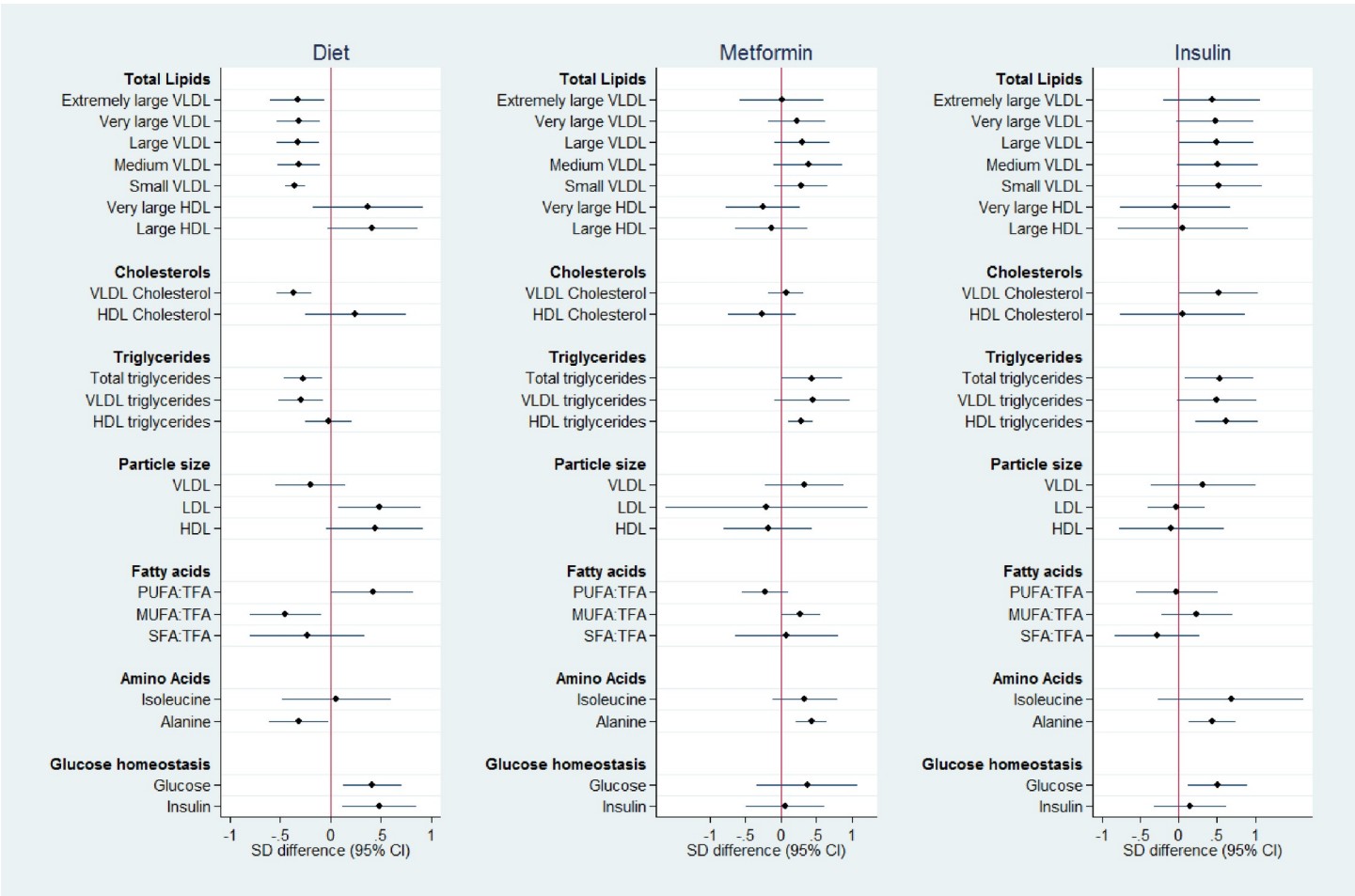

**Fig 3. Metabolite SD difference between GDM treatment groups compared to No-GDM women at time point 3, following treatment (mean 34$^{+6}$ weeks').** Data points show the standard deviation (SD) difference between treatment group and No-GDM women. Positive differences compared to No-GDM are shown to the right, negative to the left. PUFA:TFA polyunsaturated fatty acids to total fatty acid ratio, MUFA:TFA monounsaturated fatty acid to total fatty acid ratio, SFA:TFA saturated fatty acid to total fatty acid ratio.

treated groups than No-GDM women. Monounsaturated fatty acid:total fatty acid ratios (MUFA: TFA) and SFA:TFA were greater in the Metformin and Insulin Groups respectively. Isoleucine was higher in the women ultimately treated by metformin. In this fasting sample, glucose and insulin were significantly higher only in the Metformin and Insulin Groups (Fig 2, S4 Table).

*Time point 3; following treatment (random blood sample).* The mean duration of treatment, from diagnosis to blood draw at time point 3, was 7.1 weeks' (range 4.9–12, SD 0.9). As only final treatment modality was recorded, the number of weeks on this treatment was unknown. Differences in lipid species and other analytes observed at the time of the OGTT between Diet and No-GDM groups remained evident after treatment, with an additional greater difference in total VLDL lipids between these groups at the later time point. Similar trends were also maintained between the Insulin and No-GDM Groups, although treatment was associated with convergence towards the No-GDM profile. 'Normalisation' towards the No-GDM group was evident following treatment with metformin (Metformin Group), including glucose and insulin, with differences remaining only for total triglycerides and triglycerides in HDL, and alanine (Fig 3).

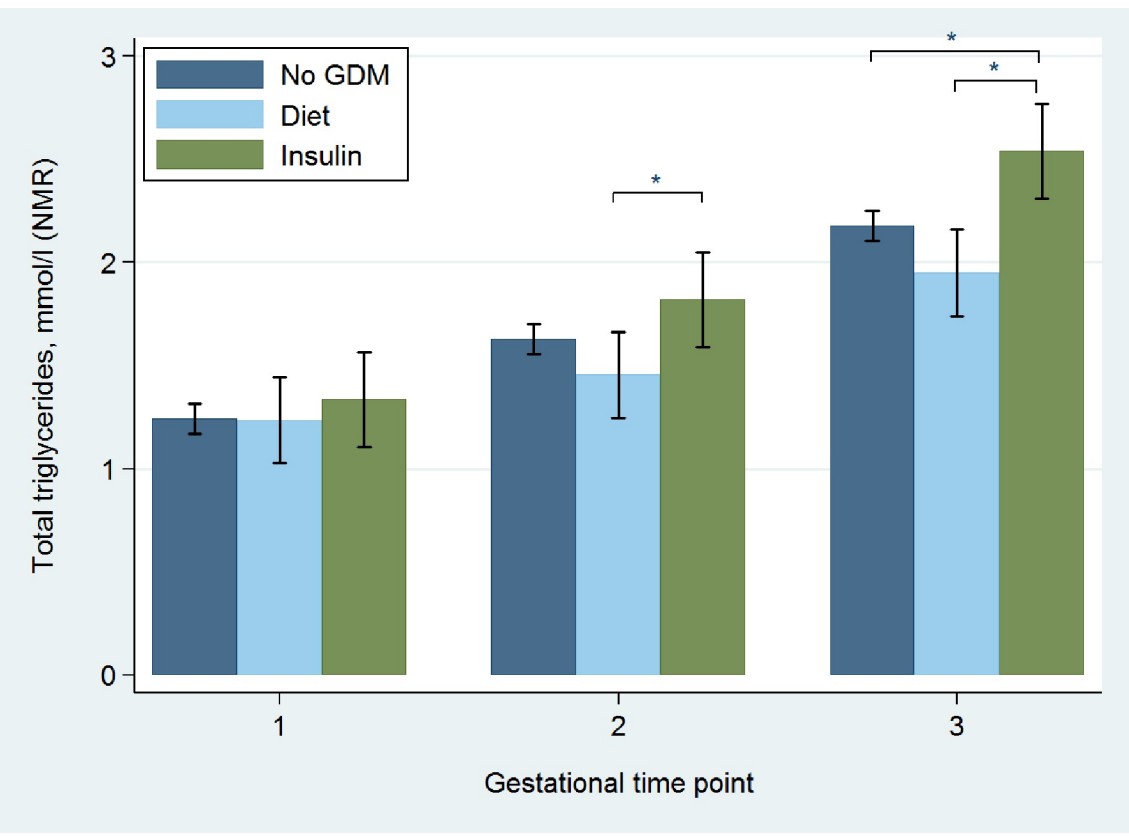

**Fig 4. Total triglyceride measurements in diet treated, insulin treated and No-GDM women at 3 gestational time points across pregnancy.** time point 2 (mean $27^{+5}$ weeks') was fasting. 95% CI, not adjusted. * p value <0.05.

*Gestational profile of glucose and total triglycerides in Insulin versus Diet Groups*. Figs 4 and 5 illustrate the gestational profile of total triglyceride and glucose concentrations of No-GDM, Diet and Insulin Groups. At time points 2 and 3 triglycerides were greater in the Insulin Group compared to the Diet-treated women, with no difference between the Diet Group and women without GDM. At time point 2 (fasting sample), glucose concentration in the Insulin Group was higher than both women treated with diet and those without GDM, with no difference between these latter groups.

## Discussion

To our knowledge there has been no previous attempt to assess the metabolic profile in GDM in obese women according to the three conventional modalities of treatment; diet, metformin and insulin. Whilst, as might be anticipated, treatment led towards convergence of analytes towards the 'norm', we also identified differing analyte profiles early in pregnancy amongst women, according to their eventual treatment regime.

### Comparison between treatment groups

The rationale for treatment of GDM with diet or a pharmacological approach is generally based on severity of hyperglycaemia, but we suggest that these clinical practices may be unintentionally predicated, at least in part, by aetiological differences. Trends in analyte concentrations were most evident from early gestation between women in whom GDM was treated with

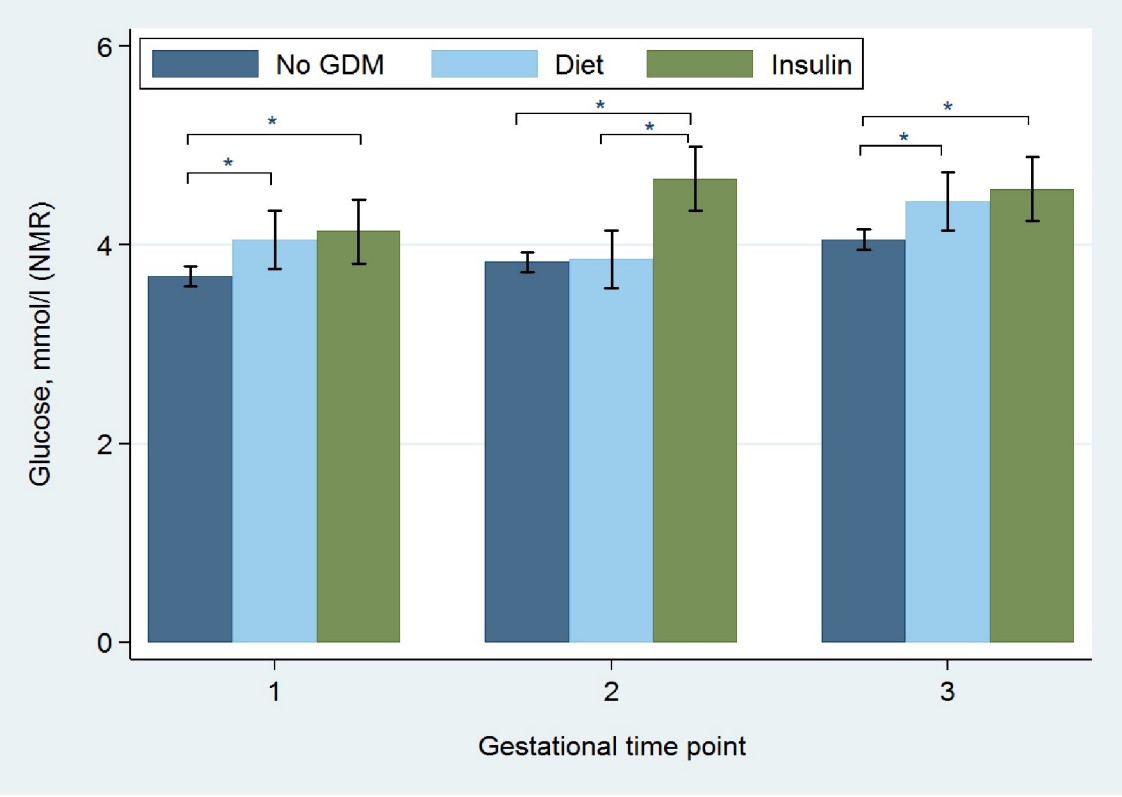

**Fig 5. Glucose measurements in diet treated, insulin treated, and No-GDM women at 3 gestational time points across pregnancy.** time point 2 (mean 27$^{+5}$ weeks') was fasting. 95% CI, not adjusted. * $p$ value <0.05.

diet and those treated with insulin; throughout pregnancy, women ultimately treated with insulin exhibited a more insulin resistant profile, whereas women whose glycaemia was ultimately controlled by diet demonstrated a markedly non-insulin resistant profile which could indicate a different pathway to GDM, possibly through insufficient secretion of insulin.

The gestational profile of insulin resistance identified from early pregnancy onwards in the Insulin and to a lesser degree, the Metformin Groups, as defined by the NMR spectrum, also included higher lipid constituents of VLDL subclasses, lower HDL constituents and smaller LDL particle size, a profile described previously in non-pregnant insulin resistant subjects [16–18]. Of potential relevance, an 'Insulin Resistance Score' based on similar indices measured by NMR is now commercially available for type 2 diabetes mellitus (T2DM) risk assessment in non-pregnant individuals [19] and a similar scoring system could be envisaged for determining GDM risk.

Similarly, the fatty acid and amino acid profiles characteristic of insulin resistance, as observed in the women with GDM, have been identified previously in non-pregnant populations, particularly higher monounsaturated and saturated fatty acids, and branched-chain amino acids [13, 20].

## Women treated for GDM *v* women without GDM

When comparing the treatment groups to women without GDM, differences in metabolite profiles were also evident from the earliest point of measurement, many weeks before treatment; the Metformin and Insulin Groups already demonstrating an 'unfavourable', more insulin resistant profile, incorporating amongst other markers, higher total lipids in VLDL

subclasses and larger VLDL particle size. Women ultimately treated with diet did not share these characteristics; total lipids in VLDLs were similar in concentration to No-GDM women, and LDL and HDL particles were larger. As expected, all three groups exhibited higher glucose concentrations than women without GDM. At the time of GDM diagnosis, the divergence in analytes between groups was particularly striking; the Diet Group now showed a more favourable lipid profile than the women without GDM, in contrast to raised insulin resistance markers in the insulin-treated group. The Metformin Group showed intermediate lipid values, although with an unfavourable fatty acid and amino acid profile. In accordance with aetiological diversity amongst groups, women treated only with metformin, and women requiring insulin were unable to maintain normoglycaemia on fasting (time point 2), whereas glucose and insulin concentrations in the Diet Group were similar to women without GDM.

## Different GDM subgroups of obese women?

The inference that the obese women treated with diet may represent a distinct subgroup is supported by a previous study inferring diverse pathways leading to GDM amongst BMI heterogeneous women. Similar differences in fasting glucose and insulin concentrations to those we describe between diet and insulin treated groups were identified in women with GDM defined by a poor insulin secretion profile (fasting glucose 76mg/dl; 72–79, fasting insulin 6.0μl/ml; 4.6–6.7) or those with an insulin resistant profile (90mg/dl; 81–94, 13.6μl/ml 9.9–20.5) respectively [2].

The consistently 'lower' insulin resistant profile in the diet treated group throughout pregnancy and following treatment, adds strength to the case for a different aetiology between groups, and we hypothesise that women in this study cohort whose GDM is treatable by diet may represent a sub group with a poor insulin secretion profile. This is supported by the observation that following treatment, glucose homeostasis remained abnormal, with relative hyperglycaemia and hyperinsulinaemia (non-fasting). In contrast, insulin resistance markers in women treated with metformin, converged towards those in women without GDM after treatment, with additional improvements in glucose and insulin concentration. Women in the Insulin Group did not achieve a similar degree of 'normalisation', but interpretation is limited as the glycaemic control achieved is unknown.

A difference in gestational age at delivery was evident between GDM treatment groups, likely reflecting the clinical approach to the timing of delivery between these groups. However, despite differing underlying pathophysiological processes and potential severity of disease, other pregnancy outcomes between treatment groups did not differ significantly (Table 1), although this may reflect the small numbers in each group.

## Strength and weaknesses

We believe there has been no previous exploration of mechanistic heterogeneity of treatment groups using metabolic profiling amongst obese women.

This is a proof of concept study involving a subgroup analysis of a large cohort; although women included were demographically similar to those not included, it is a potential weakness that data may be missing not at random (MNAR) [21].

Based on a known effect of the UPBEAT intervention on metabolite profiles [22], a decision was made *a priori* to explore subgroups in the control arm of the trial only. It is accepted that this resulted in a reduction in the number of women in the GDM treatment groups, which is a limitation of this study.

This, the first detailed description of metabolic profiles in relation to treatment in women with GDM prompts further and more detailed investigation; confirmation of phenotypic

subgroups as indicated by metabolic analyses is required amongst a larger patient sample, and different ethnic subgroups. Measurement of more specific markers of insulin secretion and sensitivity could further define pathophysiological subgroups.

The UPBEAT trial did not have a standardised protocol for GDM treatment which may have differed between centres, although analyses were clustered by centre to minimise bias. As GDM treatment modality was obtained following delivery, the time of initiation and cessation of treatment was commonly not recorded. No formal correction for multiple testing was undertaken because of the exploratory nature of the analysis and the small sample size.

In summary, targeted metabolomic analyses have suggested diverse profiles according to treatment modality. Confirmation in larger populations is required and if validated could provide a rationale for early stratification and appropriate therapy.

## Supporting information

**S1 Table. Analytical methodologies.**
(DOCX)

**S2 Table. Comparison of GDM women in treatment modality cohort compared to those excluded (control arm).**
(DOCX)

**S3 Table. Absolute analyte concentrations by treatment modality, time point 1, 10 weeks before diagnosis/treatment (mean $17^{+0}$ weeks').**
(DOCX)

**S4 Table. Absolute analyte concentrations by treatment modality, time point 2, at time of OGTT (mean $27^{+5}$ weeks').**
(DOCX)

**S5 Table. Absolute analyte concentrations by treatment modality, time point 3, following treatment (mean $34^{+6}$ weeks').**
(DOCX)

**S1 Fig. Flow diagram: Women with documented GDM treatment modality and complete biochemical data at trial time points 1 (mean $17^{+0}$ weeks'), 2 (mean $27^{+5}$ weeks') and 3 (mean $34^{+6}$ weeks') included in analyses of metabolite phenotypes by treatment modality.**
(DOCX)

## Acknowledgments

We thank staff in the UPBEAT Consortium (full list of personnel below) and participants for their time, interest and patience. We thank E. Butler and S. J. Duffus (Institute of Cardiovascular and Medical Sciences, University of Glasgow, UK) for technical support.

## UPBEAT Consortium personnel

**King's College London/Guy's and St Thomas' NHS Foundation Trust** Lucilla Poston, lead author for Consortium (lucilla.poston@kcl.ac.uk), Andrew Shennan, Annette Briley, Claire Singh, Paul Seed, Jane Sandall, Thomas Sanders, Nashita Patel, Angela Flynn, Shirlene Badger, Suzanne Barr, Bridget Holmes, Louise Goff, Clare Hunt, Judy Filmer, Jeni Fetherstone, Laura Scholtz, Hayley Tarft, Anna Lucas, Tsigerada Tekletdadik, Deborah Ricketts, Carolyn Gill,

Alex Seroge Ignatian, Catherine Boylen, Funso Adegoke, Elodie Lawley, James Butler, Rahat Maitland, Matias Vieira, Dharmintra Pasupathy.

**King's College Hospital** Eugene Oteng-Ntim, Nina Khazaezadeh, Jill Demilew, Sile O'Connor, Yvonne Evans, Susan O'Donnell, Ari de la Llera, Georgina Gutzwiller, Linda Hagg.

**Newcastle University/Newcastle NHS Foundation Trust** Stephen Robson, Ruth Bell, Louise Hayes, Tarja Kinnunen, Catherine McParlin, Nicola Miller, Alison Kimber, Jill Riches, Carly Allen, Claire Boag, Fiona Campbell, Andrea Fenn, Sarah Ritson, Alison Rennie, Robin Durkin, Gayle Gills, Roger Carr.

**Glasgow University and Greater Clyde Health Board** Scott Nelson, Naveed Sattar, Therese McSorley, Hilary Alba, Kirsteen Paterson, Janet Johnston, Suzanne Clements, Maxine Fernon, Savannah Bett, Laura Rooney, Sinead Miller, Paul Welsh, Lynn Cherry.

**Central Manchester Hospitals Foundation Trust** Melissa Whitworth, Natalie Patterson, Sarah Lee, Rachel Grimshaw, Christine Hughes, Jay Brown.

**City Hospital Sunderland** Kim Hinshaw, Gillian Campbell, Joanne Knight.

**Bradford Royal Infirmary** Diane Farrar, Vicky Jones, Gillian Butterfield, Jennifer Syson, Jennifer Eadle, Dawn Wood, Merane Todd.

**St George's NHS Trust, London** Asma Khalil, Deborah Brown, Paola Fernandez, Emma Cousins, Melody Smith.

**University College London** Jane Wardle, Helen Croker, Laura Broomfield (Weight Concern—Registered Charity.No. 1059686©)

**University of Southampton** Keith Godfrey, Sian Robinson, Sarah Canadine, Lynne Greenwood.

**Trial Steering Committee** Catherine Nelson-Piercy, Stephanie Amiel, Gail Goldberg, Daghni Rajasingham, Penny Jackson, Sara Kenyon, Patrick Catalano.

## Author Contributions

**Conceptualization:** Sara L. White, Shahina Begum, Matias C. Vieira, Deborah L. Lawlor, Naveed Sattar, Scott M. Nelson, Dharmintra Pasupathy, Lucilla Poston.

**Data curation:** Sara L. White, Paul Seed.

**Formal analysis:** Sara L. White, Shahina Begum, Paul Seed.

**Funding acquisition:** Sara L. White, Deborah L. Lawlor, Naveed Sattar, Scott M. Nelson, Lucilla Poston.

**Investigation:** Sara L. White, Paul Welsh.

**Methodology:** Sara L. White, Matias C. Vieira, Deborah L. Lawlor, Naveed Sattar, Scott M. Nelson, Dharmintra Pasupathy, Lucilla Poston.

**Supervision:** Lucilla Poston.

**Validation:** Sara L. White, Shahina Begum, Matias C. Vieira, Paul Seed, Dharmintra Pasupathy, Lucilla Poston.

**Visualization:** Sara L. White.

**Writing – original draft:** Sara L. White, Lucilla Poston.

**Writing – review & editing:** Sara L. White, Shahina Begum, Matias C. Vieira, Paul Seed, Deborah L. Lawlor, Naveed Sattar, Scott M. Nelson, Paul Welsh, Dharmintra Pasupathy, Lucilla Poston.

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
