## [Decision Letter · Decision Letter 0]

20 Nov 2019

PONE-D-19-31068

Metabolic phenotyping by treatment modality in obese women with gestational diabetes suggests diverse pathophysiology: an exploratory study

PLOS ONE

Dear Dr White,

Thank you for submitting your manuscript to PLOS ONE. After careful consideration, we feel that it has merit but does not fully meet PLOS ONE’s publication criteria as it currently stands. Therefore, we invite you to submit a revised version of the manuscript that addresses the points raised during the review process.

As detailed by the second reviewer please why the metabolites were measured in the control group and not in the intervention arm as well? in addition, the small sample size needs to be recognised as a limitation

We would appreciate receiving your revised manuscript by Jan 04 2020 11:59PM. To enhance the reproducibility of your results, we recommend that if applicable you deposit your laboratory protocols in protocols.io, where a protocol can be assigned its own identifier (DOI) such that it can be cited independently in the future. For instructions see: http://journals.plos.org/plosone/s/submission-guidelines#loc-laboratory-protocols

We look forward to receiving your revised manuscript.

Kind regards,

Stephen L Atkin, MD

Academic Editor

PLOS ONE

Journal Requirements:

4. One of the noted authors is a group or consortium: UPBEAT Consortium

In addition to naming the author group, please list the individual authors and affiliations within this group in the acknowledgments section of your manuscript. Please also indicate clearly a lead author for this group along with a contact email address.

SMN discloses receipt of lecture/other fees from Roche Diagnostics, outside the scope of this work. DAL discloses receipt of fees from Roche Diagnostics and Ferring Pharmaceuticals for research unrelated to this paper. All other authors declare that there is no declaration of interest associated with their contribution to this manuscript.

Reviewers' comments:

Reviewer's Responses to Questions

**Comments to the Author**

1. Is the manuscript technically sound, and do the data support the conclusions?

Reviewer #1: Yes

Reviewer #2: Yes

2. Has the statistical analysis been performed appropriately and rigorously? 

Reviewer #1: Yes

Reviewer #2: Yes

3. Have the authors made all data underlying the findings in their manuscript fully available?

Reviewer #1: Yes

Reviewer #2: Yes

4. Is the manuscript presented in an intelligible fashion and written in standard English?

Reviewer #1: Yes

Reviewer #2: Yes

5. Review Comments to the Author

Reviewer #1: Interesting further light on heterogeneity of GDM, supporting clinical impressions. May not influence initial treatment but provides more evidence for clinical practice, and also explanation of patterns of blood glucose abnormalities to patients i.e. raised fasting alone versus raised post prandial +/- raised fasting.

Reviewer #2: This paper is a subgroup analysis of the UPBEAT trial describing changes in some markers of insulin resistance and insulin secretion in Obese Women with and without diabetes. It is a proof of concept study.

The study methods have been published before. The authors acknowledged some of the limitations in their methods, including the absence of a unified protocol for the treatment of GDM.

The study, overall, is clearly written and is easy to follow.

I was left wondering, and perhaps many other readers will be, why the metabolites were measured in the control group and not in the intervention arm as well? I think the paper would have been more interesting if they reported on both groups.

The numsber of subjects in the GDM groups was very small, and this should be acknowledged one of the limitation.

6. PLOS authors have the option to publish the peer review history of their article (what does this mean?). If published, this will include your full peer review and any attached files.

Reviewer #1: Yes: Stephen Beer

Reviewer #2: Yes: Mohammed Bashir

---

## [Author Response · Author response to Decision Letter 0]

12 Dec 2019

Referee 2 made two comments which required a response and amendment of the manuscript. 

“I was left wondering, and perhaps many other readers will be, why the metabolites were measured in the control group and not in the intervention arm as well? I think the paper would have been more interesting if they reported on both groups. The number of subjects in the GDM groups was very small, and this should be acknowledged as one of the limitations.”

We have responded with the following addition to the discussion text:

“Based on a known effect of the UPBEAT intervention on metabolite profiles (Mills HL et al.), a decision was made a priori to explore subgroups in the control arm of the trial only. It is accepted that this resulted in a reduction in the number of women in the GDM treatment groups, which is a limitation of this study.”

---

## [Decision Letter · Decision Letter 1]

27 Feb 2020

PONE-D-19-31068R1

Metabolic phenotyping by treatment modality in obese women with gestational diabetes suggests diverse pathophysiology: an exploratory study

PLOS ONE

Dear Dr White,

Thank you for submitting your manuscript to PLOS ONE. After careful consideration, we feel that it has merit but does not fully meet PLOS ONE’s publication criteria as it currently stands. Therefore, we invite you to submit a revised version of the manuscript that addresses the points raised during the review process.

please address the statistical queries that have been raised

We would appreciate receiving your revised manuscript by Apr 12 2020 11:59PM. To enhance the reproducibility of your results, we recommend that if applicable you deposit your laboratory protocols in protocols.io, where a protocol can be assigned its own identifier (DOI) such that it can be cited independently in the future. For instructions see: http://journals.plos.org/plosone/s/submission-guidelines#loc-laboratory-protocols

We look forward to receiving your revised manuscript.

Kind regards,

Stephen L Atkin, MD

Academic Editor

PLOS ONE

Reviewers' comments:

Reviewer's Responses to Questions

**Comments to the Author**

1. If the authors have adequately addressed your comments raised in a previous round of review and you feel that this manuscript is now acceptable for publication, you may indicate that here to bypass the “Comments to the Author” section, enter your conflict of interest statement in the “Confidential to Editor” section, and submit your "Accept" recommendation.

Reviewer #3: (No Response)

2. Is the manuscript technically sound, and do the data support the conclusions?

Reviewer #3: Yes

3. Has the statistical analysis been performed appropriately and rigorously? 

Reviewer #3: Yes

4. Have the authors made all data underlying the findings in their manuscript fully available?

Reviewer #3: Yes

5. Is the manuscript presented in an intelligible fashion and written in standard English?

Reviewer #3: Yes

6. Review Comments to the Author

Reviewer #3: Table 1- some of the data is n and %, but this heading is not included. Add with mean (sd) and median (IQR).

Discussion/ stats methods: the discussion states: “Despite differing underlying pathophysiological processes and potential severity of disease, outcomes between treatment groups did not differ significantly (Table 1)” List in the manuscript the statistical methods used to determine this.

Consider showing 95% confidence intervals for the outcomes listed in table 1 to further show the lack of differences in outcome between treatment groups.

7. PLOS authors have the option to publish the peer review history of their article (what does this mean?). If published, this will include your full peer review and any attached files.

Reviewer #3: No

---

## [Author Response · Author response to Decision Letter 1]

3 Mar 2020

Reviewer 3 made three comments which required a response or amendment of the manuscript. 

1. “Table 1 – some of the data is n and %, but this heading is not included. Add with mean (sd) and median (IQR).”

We are grateful for this observation and we have updated the table as requested.

2. “Discussion/stats methods: the discussion states: ‘Despite differing underlying pathophysiological processes and potential severity of disease, outcomes between treatment groups did not differ significantly (Table 1) – list in the manuscript the statistical methods used to determine this”.

We have added the following sentence at the end of ‘Statistical Analysis’ in the Materials and Methods section (P8, L150-151):

“Pregnancy outcomes between GDM treatment groups were compared using either one-way ANOVA or the Kruskal Wallis test depending on the distribution of data.”

3. “Consider showing 95% confidence intervals for the outcomes listed in table 1 to further show the lack of differences in outcome between treatment groups.”

Thank you for this suggestion. As noted, one-way ANOVA and Kruskal Wallis were used to compare outcomes between treatment groups. These methods do not calculate 95% CI. 

4. We should like to add a comment (P17, L314-315) regarding differences in gestational age at delivery between the groups, previously omitted. We noted this difference when double checking the data for this revision – our apologies for this oversight. The difference resonates with current clinical practice and may be of interest to the reader.

---

## [Editor Report · Decision Letter 2]

6 Mar 2020

Metabolic phenotyping by treatment modality in obese women with gestational diabetes suggests diverse pathophysiology: an exploratory study

PONE-D-19-31068R2

Dear Dr. White,

We are pleased to inform you that your manuscript has been judged scientifically suitable for publication and will be formally accepted for publication once it complies with all outstanding technical requirements.

With kind regards,

Stephen L Atkin, MD

Academic Editor

PLOS ONE
---

## [Editor Report · Acceptance letter]

18 Mar 2020

PONE-D-19-31068R2 

Metabolic phenotyping by treatment modality in obese women with gestational diabetes suggests diverse pathophysiology: an exploratory study 

Dear Dr. White:

I am pleased to inform you that your manuscript has been deemed suitable for publication in PLOS ONE. Congratulations! Your manuscript is now with our production department. 

With kind regards,

on behalf of

Dr. Stephen L Atkin 

Academic Editor

PLOS ONE